# The Association between Spermidine/Spermine N^1^-Acetyltransferase (SSAT) and Human Malignancies

**DOI:** 10.3390/ijms23115926

**Published:** 2022-05-25

**Authors:** Ryan Tsz-Hei Tse, Xiaofan Ding, Christine Yim-Ping Wong, Carol Ka-Lo Cheng, Peter Ka-Fung Chiu, Chi-Fai Ng

**Affiliations:** 1S. H. Ho Urology Centre, Department of Surgery, The Chinese University of Hong Kong, Hong Kong, China; ryantse@surgery.cuhk.edu.hk (R.T.-H.T.); christinewong@surgery.cuhk.edu.hk (C.Y.-P.W.); carolcheng@surgery.cuhk.edu.hk (C.K.-L.C.); 2Department of Surgery, The Chinese University of Hong Kong, Hong Kong, China; johnding@surgery.cuhk.edu.hk

**Keywords:** SSAT, mechanisms, cancers development, polyamines

## Abstract

Spermidine/spermine N^1^-acetyltransferase (SSAT) functions as a critical enzyme in maintaining the homeostasis of polyamines, including spermine, spermidine, and putrescine, in mammalian cells. SSAT is a catalytic enzyme that indirectly regulates cellular physiologies and pathways through interaction with endogenous and exogenous polyamines. Normally, SSAT exhibits only at a low cellular level, but upon tumorigenesis, the expression, protein level, and activities of SSAT are altered. The alterations induce cellular damages, including oxidative stress, cell cycle arrest, DNA dynamics, and proliferation by influencing cellular mechanisms and signaling pathways. The expression of SSAT has been reported in various studies to be altered in different cancers, and it has been correlated with tumor development and progression. Tumor grades and stages are associated with the expression levels of SSAT. SSAT can be utilized as a target for substrate binding, and excreted metabolites may be used as a novel cancer biomarker. There is also potential for SSAT to be developed as a therapeutic target. Polyamine analogs could increase SSAT expression and increase the cytotoxicity of chemotherapy to tumor cells. Drugs targeting polyamines and SSAT expression have the potential to be developed into new cancer treatments in the future.

## 1. Introduction

Spermidine/spermine N^1^-acetyltransferase (SSAT), also known as SAT1 or SSAT-1, is a rate-limiting enzyme in mammalian cells. SSAT serves as an important homeostatic regulator in polyamine metabolic pathways to maintain a properly balanced ratio of polyamines in cells, thereby preventing an overaccumulation of polyamines that may become cytotoxic [1,2]. SSAT is usually associated with spermine, spermidine, and putrescine regarding its catalytic role (Figure 1). Back in 1973, SSAT was first observed in rat liver. It was discovered during studies on the ability of carbon tetrachloride to cause an increase in putrescine and a loss of spermidine and spermine in rodent liver. Radioactive polyamines employed in the studies showed that these changes were due to increased conversion of these higher polyamines into putrescine but without a change in an oxidase, which catalyzed this reaction. The authors discovered a cytosolic enzyme and found that acetylated polyamines were highly inducible by carbon tetrachloride, and it was suggested that the acetylated products were the true oxidase substrate. After the purification and characterization of SSAT, it was revealed that it functioned quite distinctly from other enzymes, such as histone acetylases, for which SSAT acetylated only the N^1^ position of the spermidine and both ends of the symmetrically structured spermine [2,3]. As a result, it was found that SSAT catalyzes N^1^-acetylation by adding acetyl groups to the aminopropyl end of spermidine and spermine to reduce charges on the polyamines [4]. This alters their ability to bind acidic macromolecules, thereby generating acetylated derivatives to be excreted from cells or producing substrates for oxidation [2]. Since SSAT functions as a regulator of cellular polyamine content, it indirectly governs cellular physiology, including ion channel regulation, membrane potential, intercellular pH, and cell volume. By regulating polyamine homeostasis, essential cellular pathways, and processes, such as cell proliferation, the inflammatory process, lipid metabolism, and cell death, are maintained [5,6]. The catabolism of acetylated polyamines occurs when they are oxidized by acetylpolyamine oxidase (APAO) to generate reactive oxygen and aldehyde species, such as hydrogen peroxide (H_2_O_2_) and 3-acetoaminopropanal (3-AAP), which are responsible for damaging DNA and critical cellular components [7,8]. Meanwhile, spermine oxidase (SMO), localized in the cytoplasm and nucleus of human cells, is also able to generate H_2_O_2_ and induce genetic damage by the exhaustion of spermine [9]. SSAT is regulated at multiple levels, including increased transcription and translation when cellular polyamines or polyamine analogs accumulate, whilst the degradation of SSAT proteins and the incorrect splicing of SSAT mRNA are reduced [2]. Under normal circumstances, the SSAT gene resides in the X-chromosome at location Xp22.1 and encodes SSAT [10]. SSAT contains an upstream polyamine-responsive element (PRE), which allows for the activation of SSAT transcription in high polyamine content. The onset of SSAT transcription, also involved polyamine-modulated factor-1 (PMF-1), partners with NF-E2-related factor (Nrf-2), a transcription factor that interacts constitutively with the PRE and Maf protein families [11]. Nonetheless, under oxidative stress, H_2_O_2_ upregulates SSAT transcription, which is mediated by Nrf-2 and NF-κB. This in turn accelerates polyamine degradation and generates toxic products to promote carcinogenesis [12].

Apart from normal cellular physiology, several studies have provided insights into the role of SSAT in human diseases and malignancies. In addition to the reported relationship between the overexpression of SSAT in keratosis follicularis spinulosa decalvans, a rare X-linked disease, and the under expression of SSAT in depression and suicide [13,14], SSAT was also associated with human cancers. The upregulation of SSAT induces alterations in polyamines and has been investigated in various human cancers [5,15,16,17]. SSAT exists in very small amounts in normal healthy cells; nonetheless, its levels can be increased by numerous factors, including toxic agents, hormones, drugs, and growth factors [5]. Mammalian cells overexpressing SSAT exhibited a rapid arrest in protein synthesis and cell proliferation as a result of the depletion of spermine and spermidine by the enzyme [18]. In addition, SSAT mRNA expression and protein content were found to be upregulated by approximately 5- to 10-fold in human prostate, breast, and lung primary tumor tissue when compared to normal adjacent tissue [5]. Increased polyamine levels, and their acetylated derivatives to be excreted, were detected in cell culture medium, body fluids, urine, and plasma. This could be evidence of increased SSAT activity in cancers [18]. It was observed that in cancer, such as prostate cancer, the expression of SSAT increased in order to prevent the accumulation of toxic polyamines from reaching levels that will prevent cancer cell progression [19]. In view of the role of SSAT in various human malignancies, we reviewed the alteration of SSAT levels in different cancers, the underlying mechanisms, and the therapeutic potentials of SSAT in anticancer therapies.

## 2. Involvement of SSAT in Cellular Mechanisms

The role of SSAT in different cellular pathways and mechanisms has been reported by different in vitro studies by the overexpression or knockdown of SSAT in different cell lines. It was revealed that the SSAT was involved in several cellular pathways, such as cell cycle and DNA repair, and therefore, when mutations occurred in SSAT regulation, cell migration and proliferation were implicated, which could potentially result in tumorigenesis [20].

### 2.1. Cell Cycle and DNA Damage

Some studies have revealed that SSAT is involved in the cell cycle and DNA dynamics. Thakur et al. employed the U87MG cell line to investigate the mechanistic role of SSAT in brain tumor biology. The knockdown of SSAT in stably transduced U87MG was utilized to investigate gene expression aberrations. The authors observed that genes involved in, for example, the cell cycle, mitosis, and DNA metabolism and repair were significantly affected by SSAT, thereby driving aggressive tumor biology. The authors also demonstrated the association of SSAT with maternal embryonic leucine zipper kinase (MELK) and the enhancer of zeste homolog 2 (EZH2), which are both necessary for stem cell maintenance, tumor growth, and resistance to ionizing radiation (IR) in brain cancer. Mutated SSAT was knockined in U87MG with CRISPR/Cas9, and the authors observed that the SSAT regulated the MELK and EZH2 by direct interaction with chromatin [20]. Another study in human glioma suggested that SSAT knockdown in the U87MG and D54MG cell lines could regulate BRCA1 expression and the DNA repair process. The results indicated that DNA damage was more prominent in SSAT-depleted cells when compared to the controls after ionizing radiation. The authors suspected that alterations in SSAT might affect BRCA1 through histone acetylation, and their results implied that SSAT-deficient cells exhibited decreased levels of acetylated H3 localized to the BRCA1 promoter, thereby inhibiting the homologous recombination pathway to repair DNA damage. As a result, the authors suggested that by inhibiting SSAT, brain tumors could be sensitized to radiation, which in turn increased therapeutic responses [21]. Another study conducted by Zahedi et al. illustrated the relationship of SSAT overexpression with cell cycle arrest and DNA damage and repair. The authors employed HEK-293, which is a tumorigenic yet non-malignant immortalized human kidney cell line, to demonstrate their results. They observed that cell proliferation was reduced; nonetheless, cell viability was not affected in conditional SSAT-overexpressing HEK-293 cells, which were free from ubiquitin-mediated degradation. In addition, the authors observed an increased population of G_2_-arrested and endoreduplicated cells on days 4 and 6, respectively, under the SSAT-overexpressing condition. It was suggested that the possible underlying mechanism involved the ERK signaling pathway, as phosphorylated active forms of ERK1 and ERK2 were induced. Moreover, SSAT overexpression increased the concentration of H_2_O_2_, which could potentially cause cell death and DNA damage. In terms of the DNA damage and repair pathway, their results indicated that the DNA of cells overexpressing SSAT endured a significantly higher degree of DNA damage than the controls. Nonetheless, the ATM/ATR DNA repair pathway was activated in SSAT-overexpressing HEK-293 due to the observations of increased activated Chk1 and Chk2 levels [22]. Androgen is known to play a critical role in prostate cancer development; nonetheless, mechanisms underlying androgen-induced oxidative stress generation in prostate cancer remained elusive. Mehraein-Ghomi et al. reported that a direct binding of androgen-activated androgen receptor (AR) with transcription factor JunD induced SSAT overexpression in the human prostate cancer cell line LNCaP. The results indicated that activated AR required JunD to induce the transcriptional activity of the SSAT promoter. The activated AR-JunD complex was bound to an AP-1 DNA-binding sequence in the SSAT promoter to activate SSAT gene transcription. Upon androgen treatment, the vector control LNCaP exhibited a significant, >16-fold increase in the SSAT promoter activity compared to the untreated cells. The increased SSAT transcription thus increased the H_2_O_2_ production and ROS generation in the prostate cells [23].

### 2.2. Cell Proliferation, Invasion, and Migration

In addition to the cell cycle and DNA repair, SSAT was also suggested to be involved in tumor cell proliferation, invasion, and migration. Several in vitro studies on hepatocellular carcinoma (HCC) and colorectal cancer (CRC) have revealed the underlying mechanisms. Wang et al. reported the role of SSAT in transfected HCC cell lines (HepG2, SMMC7721, and Bel7402) and CRC cell lines (HCT116 and HT-29). The authors observed SSAT overexpression in SSMC7721-, HepG2-, and HCT116-inhibited colony formation and cancer cell migration and invasion, whilst the knockdown of SSAT in Bel7402 and HT-29 demonstrated a significant increase in cell proliferation, migration, and invasion. The major event that occurred after the aberration of SSAT was the alteration in the Akt/GSK3β/β-catenin signaling pathway, in particular, the phosphorylation status of GSK3β Ser9. In addition, the expression and translocation of β-catenin into the nucleus could activate downstream target genes, including c-myc, cyclin-D1, and ZEB1, to regulate cell proliferation, migration, invasion, and metastasis [24]. Sun et al. also conducted experiments to elucidate the mechanistic role of SSAT in CRC. The authors employed CRC cell lines, HT-29 and LoVo, and constructed a recombinant adenovirus to overexpress SSAT. The results demonstrated that high SSAT levels inhibited cell growth by cell cycle arrest in the S phase by suppressing the cell cycle controller cyclin A and its upstream regulator E2F-1 expression [25].

### 2.3. Oncogenic Pathway

SSAT was also identified as a transcription target of the p53 protein [26]. It is well known that the p53 gene functions as a tumor suppressor and the translated p53 protein plays a central role in preventing malignant formation [27]. Ou et al. demonstrated that the expression of SSAT was upregulated by p53 utilizing different cancer cell lines by treating the cells with Nutlin or a DNA-damaging drug. Of note, the authors suggested that the p53-null cell line H1299 demonstrated no apparent change in SSAT mRNA expression. Moreover, the authors attempted to induce p53 expression with the addition of tetracycline and overexpress p53 by transient infection of the H1299 cell lines. The results revealed that the SSAT mRNA level was increased, indicating that SSAT was a transcriptional target of p53 [26]. It was also suggested that SSAT participates in the p53-mediated ferroptotic response towards ROS stress. Studies have indicated that after SSAT is upregulated by p53, arachidonate 15-lipoxygenase (ALOX15), a member of the lipoxygenase family that is specifically responsible for oxidative-stress cell death, is induced. This ultimately sensitizes the cells to ferroptosis in the presence of ROS, which is manifested as tumor suppression in xenograft models. Of note, ferroptosis, which denotes an iron-dependent, nonapoptotic mode of cell death characterized by an accumulation of lipid ROS at the cell membrane, was observed in several lung diseases, such as lung cancer. In addition, the generation of oxidative stress as a result of SSAT overexpression, induced lipid peroxidation, and cell death have also been reported in several studies [8,9,26,28].

## 3. SSAT Alterations in Human Malignancies

The dysregulation of SSAT expression and protein levels have been reported in different human malignancies. High levels of SSAT expression were measured in human immortalized cell lines, such as human lung cancer cell line A549, human prostate cancer cell line LNCaP, and human breast tumor cell line T-47D, as well as in patient-derived breast, prostate, and lung tumor tissues [5]. SSAT promoted the polyamine catabolic pathway that could decrease intracellular spermine and spermidine levels, which are important free radical scavengers. A reduced concentration of free radical scavengers might promote chronic inflammation and initiate cancer development [29]. Several studies have also utilized a xenograft model to observe the effects of SSAT overexpression in malignancy development and progression. Notwithstanding, alterations in the expression of SSAT were seemingly dependent on different tumor types. In addition, SSAT underwent extensive post-transcriptional and translational regulations. As a result, different studies have reported discrepant patterns of SSAT dysregulation in human malignancies (Table 1).

### 3.1. Skin Cancer

As early as 2002, a skin cancer K6-SSAT transgenic mice model was established with the overexpression of the epidermal SSAT protein. Upon the induction of the mutagenic tumor initiator 7,12-dimethylbenz(a)anthracene (DMBA) and the phorbol ester tumor promoter, 12-O-tetradecanoylphorbol 13-acetate (TPA), the K6-SSAT transgenic mice developed a higher tumor burden over their normal counterparts. In addition, the tumors in the transgenic population were considerably larger than those in the non-transgenic littermates, and a significant number of the tumors progressed to carcinomas. The results indicated that 16 out of 42 tumors from the K6-SSAT treatment group were identified as carcinomas using gross appearance and histological analysis as the scoring criteria; otherwise, the incidence of carcinoma in non-transgenic mice was low and no carcinomas were observed in the tumor promotion-resistant C57BL/6 strain within <27 weeks of promotion [1]. In this study, tumor samples from transgenic mice showed marked elevations in SSAT enzyme activity and SSAT protein levels compared to tumors from non-transgenic animals. The results suggested that elevated SSAT could enhance the rate of progression from a pre-malignant to a more advanced tumor phenotype in skin tumorigenesis, and these tumors are much more likely to convert to squamous cell carcinoma [1,30].

### 3.2. Leukemia

Pirnes-Karhu et al. studied the role of SSAT in acute myeloid leukemia (AML), chronic myeloid leukemia (CML), and acute lymphoid leukemia (ALL) patients. Peripheral blood mononuclear cells (PBMC) were isolated from the patients and controls. The results indicated that the PBMCs of the AML, CML, and ALL patients exhibited significantly higher SSAT activities than those of the controls. Their xenograft models also revealed that SSAT-overexpressing mice were subjected to an increased proportion of leukocytes and a distorted proportion of lymphocytes and neutrophils in their peripheral blood compared to their wide-type littermates, which was evidence of overexpressed SSAT-promoted myeloproliferative phenotypes in mice. In addition, epigenetic factors were found to be altered in SSAT mice bone marrow cells. The authors observed that the methylation and acetylation were increased in several lysine residues when compared to control mice. Such alterations in epigenetic factors might contribute to the hematopoietic phenotype of SSAT mice [31].

### 3.3. Breast Cancer

The gene expression and enzymatic activity of SSAT were evaluated in human breast cancer. It was reported that SSAT was upregulated in the human breast cancer cell line T-47D. In addition, patient-derived breast tumors revealed approximately a 5- to 10-fold increase in SSAT mRNA levels as well as the SSAT protein contents as compared to noncancerous human epithelial cells [5]. Cervelli et al. discovered that SSAT transcription was overexpressed in tumor tissues when compared to non-tumor controls. In addition, SSAT activity was also significantly higher in malignant samples. The authors also proposed that SSAT interplays with N^1^-acetylpolyamine oxidase and gives rise to the extensive production of H_2_O_2_ locally [32]. In addition, Wallace et al. narrated similar results; intriguingly, SSAT activity in histological grade 2 tumor tissues exhibited no significant difference when compared to normal tissues. Nonetheless, tumor samples from grade 3 tumors demonstrated significantly higher SSAT activity than normal tissues. Therefore, the authors suggested that increased SSAT activity was correlated with poor prognosis [33].

### 3.4. Brain Tumor

The clinical relevance of SSAT in brain cancer has received elucidation in recent years. Studies have reported that SSAT was significantly overexpressed in high-grade glioblastoma as compared to low-grade gliomas of all subtypes. Moreover, SSAT expression was significantly correlated with poor outcome in glioblastoma and low-grade glioma cohorts. A two-fold increase in the expression of SSAT was significantly correlated with a decrease in survival time. In addition, within low-grade gliomas, the tumors with high SSAT expression performed significantly worse than low-SSAT tumors. The expression of SSAT target genes was also assessed; intriguingly, their expression was independent of subtype and distinctly higher in more aggressive gliomas [20,21].

### 3.5. Prostate Cancer

The human prostate cancer cell line LNCaP and human-derived prostate tumors demonstrated higher SSAT mRNA expression and protein content when compared to noncancerous controls [5]. Bettuzzi et al. compared the SSAT mRNA levels in a less aggressive low-grade tumor and an actively proliferating high-grade prostate cancer with Northern blot. The results indicated that there was only a minor increase in the mature SSAT mRNA form in the low-grade tumor, yet the expression of SSAT was dramatically increased in higher-grade tumors. In addition, the authors observed that SSAT was overexpressed in locally invasive prostate cancer patients with PSAs larger than 10 ng/mL and patients with poor prognosis (i.e., PSA increase after surgery, lymph node involvement, and metastasis) [19]. Huang et al. also compared SSAT mRNA levels in prostate epithelial cells under malignant, diseased (benign prostatic hyperplasia, BPH), and benign conditions. The results indicated that SSAT expression was significantly higher in prostate cancer as compared to normal prostate tissue, whilst there was no significant difference in the SSAT mRNA levels between normal prostate and tissues with BPH changes. Of note, the SSAT mRNA levels were not significantly different when comparing localized and aggressive prostate cancer, as well as when comparing prostate cancer and BPH. Nonetheless, metastatic prostate cancer tissues exhibited significantly higher SSAT mRNA levels when compared to all other groups. Intriguingly, SSAT protein levels were also high in tissues obtained from patients who ultimately progressed to advanced metastatic disease. Since the increase in the SSAT level was more pronounced in metastatic prostate cancer, SSAT was postulated to be associated with cancer progression and invasiveness. The authors also suggested that high ROS levels in adenocarcinomas, such as prostate cancer, would activate nuclear transcription factor-κB, which in turn induces SSAT expression as an underlying mechanism [34].

## 4. Therapeutic Potentials of SSAT

SSAT expression is aberrated in human cancers and may be associated with cancer development, maintenance, and progression. Therefore, the catabolic enzyme was studied as a target as a diagnostic marker and therapeutic agent. Tremendous works have been conducted in the past decades to elucidate the potential of translating laboratory results into clinical use.

### 4.1. Cancer Biomarker

Given that SSAT is associated with human malignancies and that alterations in its expression level are observed in various cancers, the substrates and/or reaction products of SSAT can be utilized as biomarkers in human cancers. Amantadine is an FDA-approved antiviral drug that complementarily binds to SSAT and undergoes acetylation to produce N-acetylamantadine (AA), whilst AA will be excreted without further metabolism. Therefore, AA could be used to visualize the elevated SSAT expression in human excrement, such as urine (Figure 2A). It was reported that the concentration of urinary AA is lower in healthy individuals than in cancer patients, including prostate, breast, and lung cancers. Intriguingly, the concentration of urinary AA exhibited a gradual increase upon progressing cancer severity, from stages II to IV [5]. Further investigation on the utilization of urinary AA as a novel cancer biomarker in breast and lung cancers was conducted. The results indicated that higher AA concentrations were observed in urine samples from patients with stage III to IV tumors and with squamous and adenocarcinoma lung cancer. It was also suggested that 0–2 post-amantadine yielded the highest AUCs of 0.689 and 0.717 in lung and breast cancers, respectively [35]. The above results indicate that the detection of AA in urine samples is non-invasive, simple, inexpensive, and can be readily carried out; therefore, SSAT can be used as a target in the development of novel cancer biomarkers and cancer differentiation.

### 4.2. Non-Steroidal Anti-Inflammatory Drug Target

Studies have indicated that non-steroidal anti-inflammatory drugs (NSAIDs) exert anti-tumorigenic effects against different cancers. Among all, NSAIDs were shown to exhibit chemopreventive effects against CRC by affecting the polyamine metabolisms. Babbar et al. revealed that sulindac sulfone induced SSAT transcription, thereby overexpressing SSAT and its enzymatic activity in the human CRC cell line Caco-2. In addition, upon treatment with sulindac sulfone, polyamines such as spermine and spermidine were shown to deplete intracellularly. The authors observed that such depletion was associated with the growth inhibition and apoptosis of Caco-2 [36]. On the other hand, sulindac was administered with difluoromethylornithine (DFMO) to prevent sporadic colorectal adenomas in a randomized controlled trial. Reductions in the recurrence and emergence of advanced and multiple adenomas were noticed [37]. In addition, Indomethacin, an NSAID approved for the treatment of pain, could also induce SSAT expression in human non-small cell lung cancer cell lines A549 and H1299. Similar to the results from Babbar et al., Indomethacin was found to exert metabolic effects on cancer cells by impairing the polyamine and amino acid metabolisms [38]. The enhanced acetylation of polyamines by SSAT and their efflux from cells were suggested to be underlying mechanisms in impaired polyamine metabolism [39]. The above studies indicate that SSAT could be a potential target of NSAIDs to exert anti-tumoral effects by disrupting polyamine homeostasis.

### 4.3. Polyamine Analogue

A well-known polyamine analog, N^1^,N^11^-diethylnorspermine (DENSpm), also known as BENSpm or BE333, was employed in clinical trials to superinduce SSAT (Figure 2B). The induction of SSAT has been shown to be associated with cytotoxic response in different human malignancies. Gabrielson et al. exposed breast cancer tissues to DENSpm and observed the induction of the SSAT protein in most tumor samples. Several phase II clinical trials with DENSpm in breast cancer demonstrated good toleration, though minimal responses were observed in subjects. Therefore, the authors suggested that the dosing schedule could be improved, as their results indicated that a polyamine catabolic response was a common event upon drug intervention [40]. In brain cancer, immortalized human glioblastoma cell lines LN229 and U87 were employed to study the effects of DENSpm in vitro. The results indicate that SSAT mRNA was overexpressed and cell detachment was observed upon drug treatment. In addition, the overexpression of SSAT induced the degradation of anti-apoptotic and adhesion-related proteins, such as Akt, mTOR, integrin α5, and integrin β1 [41]. Two phase I clinical trials attempted to treat non-small-cell lung cancer and hepatocellular carcinoma patients with DENSpm. The results indicated that DENSpm could be administered safely, although no complete or partial responses were observed in either study. Of note, stable diseases were achieved in patients upon DENSpm intervention in both trials [42,43]. A second generation of DENSpm, PG-11047, was formulated by conformationally restricting a central cis double bond at the N-terminal ethyl groups. Such a modification was intended to decrease the off-target effects in clinical trials. PG-111047 was also adopted in several clinical trials and the results indicated that a third of the patients who were diagnosed with advanced, refractory solid tumors had stable disease thereafter. On the other hand, PG-11047 could be administered at a high maximum tolerated dose than the previous generation of polyamine analogs. Intriguingly, PG-11047 was recently incorporated into self-immolative nanoparticles, and this permitted the simultaneous delivery of therapeutic nucleic acids, such as microRNA, alongside the polyamine analog [44].

In addition to being administered alone, DENSpm can be combined with other agents to exert its anti-tumor activities. Choi et al. investigated the synergistic anti-tumor effects of 5-Fluorouracil (5-FU) and DENSpm in colon carcinoma cells. The thymidylate synthase inhibitor 5-FU is commonly used as chemotherapy in colorectal carcinomas (CRC). Since polyamine metabolism was a common event upon 5-FU administration, the combination with a drug that specifically targets polyamine metabolism, such as DENSpm, was examined in vitro with the human CRC cell line HCT116. The results indicated that a strong anti-tumor synergistic effect was observed in terms of markedly decreased cell viability and increased apoptotic sub-G_1_ population [45]. In addition, Allen et al. observed similar results. The combination of SSAT and the chemotherapeutic agent induced the cell death of HCT116 as elaborated by flow cytometry and cell viability assays. In addition, DENSpm sensitized both sensitive and resistant cells to chemotherapeutic agents [46]. Strikingly, the synergism outweighed that of 5-FU with difluoromethylornithine (DFMO), which is a well-known chemoprevention agent in human malignancies [47]. There was a marked increase in the expression of SSAT in cells treated with 5-FU and DENSpm simultaneously, which led to H_2_O_2_ production followed by apoptosis in a caspase 9-dependent manner. Moreover, several physiologies changed upon 5-FU/DENSpm treatment, including losing mitochondria potential and cytochrome c release [45]. Platinum drugs are important in modern chemotherapy regimens, and a third-generation platinum drug, oxaliplatin, demonstrates anti-tumor activity. Hector et al. and Tummala et al. reported the induction of SSAT-sensitized tumor cells towards platinum drugs, and therefore, the synergism of oxaliplatin and DENSpm was examined. The authors observed that concurrent exposure to oxaliplatin and DENSpm increased SSAT mRNA expression and enzymatic activity in the human ovarian cancer cell line A2780. Regarding the cell growth inhibitory effects, treatment with DENSpm followed by oxaliplatin revealed augmented growth inhibition in A2780 cells as compared to DENSpm alone. The authors therefore suggested that the induction of SSAT by DENSpm sensitized malignant cells towards oxaliplatin. Moreover, their results depicted that the concurrent administration of oxaliplatin/DENSpm further produced a larger extent of sensitization. Of note, the oxaliplatin of concentrations that were not growth-inhibitory became cytotoxic to cells under co-exposure to both drugs [48,49]. The aforementioned second-generation DENSpm, PG-11047, was also administered together with chemotherapeutic agents in patients with advanced refractory metastatic solid tumors or lymphoma in a phase Ib clinical trial. It was reported that the dose-limiting toxicity was low upon the simultaneous administration of PG-11047 and bevacizumab, erlotinib, cisplatin, 5-fluorouracil/leucovorin, or sunitinib. Stable diseases, partial responses, and reductions in the sum of the longest diameter of baseline target lesions were observed in the patients. Therefore, PG-11047 can be safely administered alongside chemotherapeutic regimes and exerted anti-tumoral activities [50]. The above studies indicate that DENSpm alone or in synergism with other chemotherapy agents induced cell growth inhibition and apoptosis, whilst the determinant of such cytotoxic response and underlying mechanism could be the superinduction of SSAT mRNA expression and its enzymatic activity.

## 5. Conclusions

The catabolic enzyme SSAT governs polyamine homeostasis in mammalian cells. Although SSAT maintains cellular physiology and regulates vital pathways, studies have reported that the enzyme is involved in human malignancies of different origins, such as breast, prostate, brain, liver, and colorectal. Aberrations in the expression of SSAT, either overexpression or downregulation, and its enzymatic activities affect cellular proliferation and DNA repair and induce oxidative stress. It was also reported that oncogenic pathways, for instance, p53, incorporated with SSAT induce programmed cell death. Therefore, alterations in SSAT levels contribute to tumor growth, maintenance, and invasion. The feasibility of developing novel cancer biomarkers by detecting specific molecules that bind to SSAT followed by excretion demonstrated high sensitivity and specificity. Several drugs were also demonstrated to target SSAT as an anti-tumorigenic approach. Furthermore, polyamine analogs can superinduce SSAT expression and increase the cytotoxicity of chemotherapy to tumor cells.

## Figures and Tables

**Figure 1 ijms-23-05926-f001:**
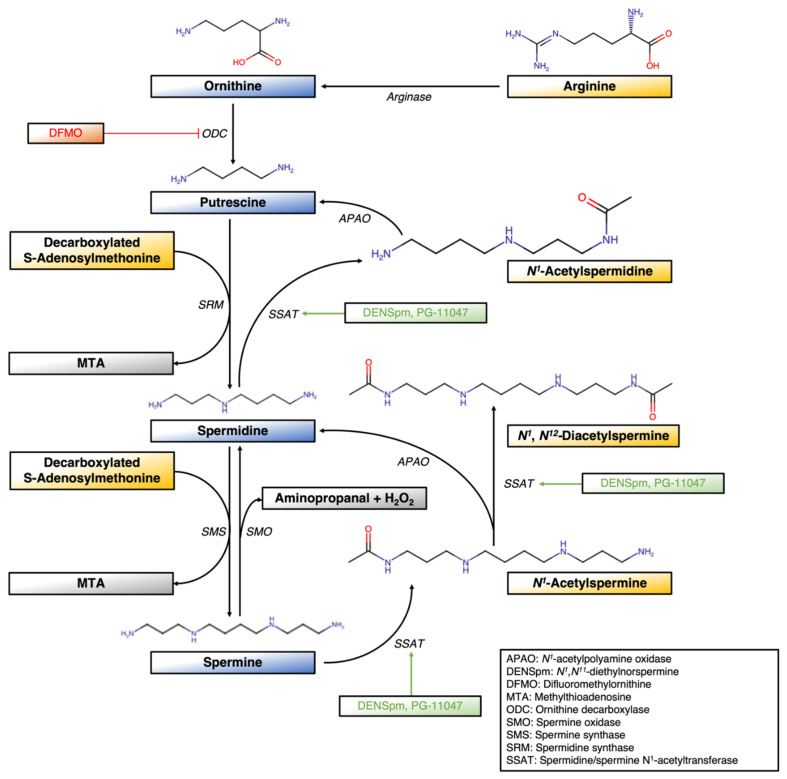
Schematic of polyamine metabolism and related polyamine analog interactions in the pathway. DMFO inhibits ODC whilst DENSpm and PG-11047 induce SSAT expression.

**Figure 2 ijms-23-05926-f002:**
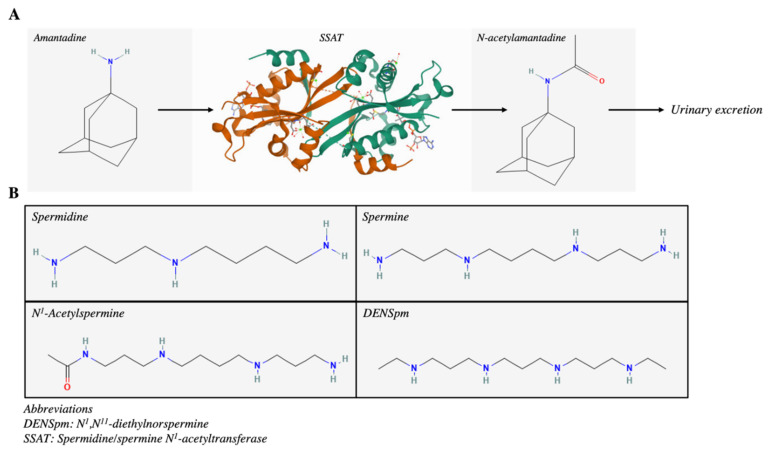
Therapeutic potential of SSAT as a biomarker and target of polyamine analogs. (**A**) Conversion of amantadine to N-acetylamantadine by SSAT, followed by urinary excretion. (**B**) Common targets of SSAT; DENSpm resembles the structure of other SSAT substrates and resulted in superinduction of SSAT.

**Table 1 ijms-23-05926-t001:** Summative table of different in vitro studies on cellular impacts induced by different SSAT levels.

SSAT Level	Cell Lines	Organ Origin	Impact	References
Overexpression	HEK-293	Kidney (Healthy)	Induced cell cycle arrest, increased cell death, increased DNA damage, activated DNA repair	[22]
	LNCaP	Prostate (PCa)	Increased H_2_O_2_ production, increased ROS production	[23]
	HepG2, SMMC7721	Liver (HCC)	Inhibited colony formation, inhibited cell migration, inhibited cell invasion	[24]
	HCT116	Colon (CRC)	Inhibited colony formation, inhibited cell migration, inhibited cell invasion	[24]
	HT-29, LoVo	Colon (CRC)	Inhibited cell growth, induced cell cycle arrest	[25]
Induction	H1299	Lung (NSCLC)	Upregulated p53 expression, increased ROS production, induced ferroptosis	[26]
Knockdown	U87MG	Brain (Glioblastoma)	Overexpression of genes involved in cell cycle, mitosis, DNA metabolism, DNA repair	[20]
	U87MG, D54MG	Brain (Glioblastoma)	Histone H3 acetylation, homologous recombination, increased DNA damage	[21]
	Bel7402	Liver (HCC)	Increased cell proliferation, increased cell migration, increased cell invasion	[24]
	HT-29	Colon (CRC)	Increased cell proliferation, increased cell migration, increased cell invasion	[24]

Abbreviations: PCa, prostate cancer; HCC, hepatocellular carcinoma; CRC, colorectal cancer; NSCLC, non-small-cell lung carcinoma.

## Data Availability

Not applicable.

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
