# Peer review of "The Association between Spermidine/Spermine N1-Acetyltransferase (SSAT) and Human Malignancies"

_ijms, 2022, doi:10.3390/ijms23115926_

Round 1

Reviewer 1 Report

The review titled "The Association between SSAT and Human Malignancies," by Tse and colleagues, covers an important topic in a field that is gaining attention. The review is well organized and covers most of the recent advances in the field. I do have a few suggestions that I believe will improve the manuscript and clarify potential misconceptions the reader may have.

  1. My main concern is that the potential role of the SSAT/PAOX pathway in producing damaging ROS and aldehydes be more clearly discussed (lines 53-55 and 140-141 in particular). Before the discovery of spermine oxidase, PAOX was often described as a source of toxic aldehydes and ROS, based on studies using the inhibitor MDL72527. It is now known, however, that this inhibitor also inhibits SMOX, and that in many instances, SMOX is the true source of DNA-damaging H2O2 and aldehydes, partially due to its cellular localization in the nucleus, while PAOX localizes to the peroxisome, where H2O2 can be rapidly neutralized by catalase. As such, caution should be used when stating that the SSAT/PAOX pathway produces DNA-damaging byproducts, as many studies did not rule out or were unaware of the contribution of SMOX (which can be induced by many of the same molecules that induce SSAT). Additionally in this context, it might be relevant to include a discussion of the DNA-shielding, free radical scavenging properties of spermidine and spermine, which would be reduced with their increased catabolism by SSAT.
  2. Lines 47-48: clarify that acetylation allows excretion, but is not always followed by excretion. N1-acetylation also produces the substrate for back-conversion via oxidation.
  3. The Introduction mostly references other review articles rather than original works. For some of the more specific descriptions, the original studies could be referenced.
  4. A brief introduction to the dysregulation of polyamine levels and their metabolism in addition to SSAT might be useful for readers outside of the field.
  5. line 118 regarding HEK-293 cells - "tumorigenic yet malignant" I believe should be "tumorigenic yet non-malignant"
  6. lines 121 and 130: HEK-293 is transposed to "239"
  7. line 161: reference is needed for SSAT as a transcriptional target of p53
  8. lines 177-178: the meaning of this sentence and how it relates to the cited references is unclear.
  9. lines 181-183: some investigators have reported downregulation of SSAT (Ou et al.) in cancer types. Perhaps SSAT is tumor-type or context dependent. The extensive post-transcriptional and translational regulation of SSAT is also important to note when interpreting gene expression levels and could be a cause of disparities.
  10. line 221-222: sentence starting with "thus the SSAT" is unclear.
  11. Table 1: I suggest indicating whether the "impact" is increased or decreased.
  12. line 274: "substrate and/or reaction product"
  13. line 299: these studies have led to clinical chemoprevention trials combing sulindac with DFMO, an inihbitor of polyamine biosynthesis, in patients at risk of colorectal cancer. These should be mentioned.
  14. section 4.3: a second generation version of DENSpm has been developed (PG-11047) that maintains the ability to highly induce SSAT while toxic off-target effects are reduced. You may also wish to mention the formulation of DENSpm and PG-11047  into nanoparticles for enhanced drug delivery and dual targeting of polyamine metabolism with a therapeutic nucleic acid. 

    DOI: 10.1007/s00280-020-04082-4

  15. paragraph starting on line 326: The study by Allen et al. 2007 should be included in combination studies. Results of a Phase Ib clinical study targeting SSAT with common chemotherapeutic agents is also included below and may add to your discussion.

    DOI: 10.1007/s00280-020-04201-1

    DOI: 6/1/128 [pii]

    10.1158/1535-7163.MCT-06-0303

Reviewer 2 Report

The authors have presented an interesting review concerning SSAT involvement in human malignancy. The presentation is well organized and presented and my suggestion is to be accepted after minor modification:

a scheme of polyamines metabolism and SSAT involvement will be helpful to all readers to better understand SSAT function and  the importance of its inhibition  in some types of cancer treatment

Round 2

Reviewer 1 Report

This revised version of the manuscript has adequately addressed my concerns. Thank you.

Reviewer 2 Report

The manuscript has been improved and I agree to be accepted. With kind regargs George